# The influence of role awareness, empathy induction and trait empathy on dictator game giving

**Kaisa Herne** [ID][1][☯]*, **Jari K. Hietanen**[2☯], **Olli Lappalainen**[2☯¤], **Esa Palosaari**[2☯]

1 Tampere University, Faculty of Management and Business, Politics, Tampere, Finland, 2 Tampere University, Faculty of Social Sciences, Pyschology, Tampere, Finland

☯ These authors contributed equally to this work.
¤ Current address: Department of Economics, Turku School of Economics, University of Turku, Turku, Finland
* kaisa.herne@tuni.fi

**Data Availability Statement:** All relevant data have been uploaded to Open Science Framework: https://osf.io/5aguw/.

**Funding:** KH, Strategic Research Council of the Academy of Finland (grant numbers 312671/

## Abstract

We ask how state empathy, trait empathy, and role awareness influence dictator game giving in a monetarily incentivized experiment. We manipulated two factors: *role awareness* (role certainty vs. role uncertainty) and *state empathy induction* (no empathy induction vs. empathy induction). Under role uncertainty, participants did not know their role as a dictator or a recipient when making their choices. State empathy was induced by asking the dictators to consider what the recipient would feel when learning about the decision. Each participant was randomly assigned into one of the four conditions, and in each condition, participants were randomly assigned into dictator and receiver roles. The role assignment took place before or after decisions were made, depending on the condition. We also studied the direct influence of *trait empathy* on dictator game giving as well as its interaction with the experimental manipulations. Trait empathy was measured by the Interpersonal Reactivity Index (IRI) and the Questionnaire of Cognitive and Affective Empathy (QCAE) before the experiment. Of our experimental manipulations, role awareness had an effect on dictator game giving; participants donated more under role uncertainty than under role certainty. Instead, we did not observe an effect of state empathy induction. Of trait empathy subscales, only affective empathy was positively associated with dictator game giving. Finally, role awareness did not influence all participants similarly but had a larger impact on those with low scores on trait empathic concern or trait affective empathy. Our results indicate that specific measures to induce altruistic sharing can be effective but their effect may vary depending on certain personal characteristics.

## Introduction

Many resource allocation situations are *asymmetric* with an influential decision-maker and a powerless recipient whose interests are at stake but who has no say over the decision. Examples include donations to charities [1] and intergenerational decisions [2]. A crucial feature of such

312675/326662) https://www.aka.fi/en/strategic-research/ The funders had no role in study design, data collection and analysis, decision to publish, or preparation of the manuscript.

**Competing interests:** The authors have declared that no competing interests exist.

decisions is that the recipient cannot show reciprocity. The dictator game is used widely to model asymmetric resource allocation [3]. In the game, a randomly selected "dictator" makes an allocation decision between him or herself and a recipient. The recipient has no say over the decision but must accept whatever the dictator offers him or her.

Giving resources to anonymous others in the dictator game matches definitions of prosocial behavior and altruism [4]. We define *altruistic behavior* as giving resources to others with a cost to oneself but without an expectation of reciprocity. Altruism does not necessarily involve a psychological motivation to help others [5]. Such behavior is in many cases socially desirable because it increases the well-being of the receiver, and as a matter of fact, may also increase the well-being of the person who acts altruistically [6]. Using this definition, altruism is one type of pro-social behavior, that is, acting in the benefit of others [5]. The empathy-altruism hypothesis states that empathy increases altruism [7–9] implying that empathy should enhance dictator game giving. It is noteworthy that we can study altruistic *behavior* with the dictator game, whereas dictators' *motivations* for their choices are not revealed in the game.

Our study covers two types of *empathy*: empathy as a temporary *state* and empathy as a more permanent *trait*. We ask first, whether two experimental manipulations, *role awareness* and *state empathy induction*, increase allocations to recipients in the dictator game. Under role uncertainty, participants make decisions before they are allocated to the dictator or recipient roles, and under state empathy induction, dictators are prompted to consider their decision from the recipient's perspective. Furthermore, we study which type of people are likely to share money in the dictator game, and which type of people are likely to be influenced by the methods that aim to increase dictator game giving. We therefore examine the direct effects of different trait empathy dimensions on dictator game giving as well as their interaction with role awareness and state empathy induction. We ask whether the influence of role awareness or state empathy induction on dictator game giving is *conditional on the decision-maker's level on one or several trait empathy dimensions*. If an interaction effect is observed, it would suggest using different methods to induce resource sharing among different types of people.

## Role awareness and dictator game behavior

In the standard dictator game, the dictator's decision is implemented with certainty. In this case, inequality aversion, social norms, altruism and social desirability may motivate the dictator to give money to the recipient. On average, dictators tend to share about 30% of their endowment but sharing varies widely depending on the specific formulation of the dictator's choice task [10, 11]. For example, dictators usually share a larger part of their endowment when the recipient is considered deserving or needy, whereas the dictator's experience of having earned the endowment reduces allocations to the recipient [10].

*Role uncertainty* means that participants do not know whether they will be dictators or recipients when they make their choices [12]. Once choices have been made, participants are randomly allocated into these two roles and only dictators' choices are implemented. Under role uncertainty, pure payoff maximizing implies zero allocations to the recipient. Existing evidence nevertheless shows that role uncertainty increases money allocations to the recipient [12]. This is somewhat puzzling because role uncertainty has a parallel incentive structure with the standard dictator game under certainty. The reason for sharing can be that role uncertainty directs attention to both roles which increases sharing of resources. Misunderstanding of the experimental instructions does not seem to account for sharing under role uncertainty [12].

It is noteworthy that role uncertainty is different from a dictator game played behind a veil of ignorance. In that case, each participant makes a dictator decision after which the dictator's

or the receiver's payoff is randomly selected to each player. This game type increases the equality of allocations, and participants' decisions behind the veil reflect their risk attitudes [13, 14].

## Trait and state empathy

Despite a variety of precise definitions [15], empathy can be characterized roughly as a multidimensional concept comprising distinct but related cognitive and affective processes, a distinction also seen in separate but interacting neural networks [16–21]. *Cognitive empathy* is a capacity to understand the emotional states of others, and *affective empathy* is a capacity to be sensitive to, and to vicariously *experience* the feelings of others [22]. E*mpathic concern*, or sympathy, is an other-oriented emotional response congruent with the perceived welfare of the other person [4, 22]. Empathic concern (feeling *for*) can be separated from affective empathy (feeling *as*) because affective empathy does not necessarily include compassion and caring [23].

The empathy-altruism hypothesis states that empathy evokes a motivation to help [7–9]. Brain imaging studies have shown that activation of the neural systems associated with affective empathy is associated with dictator game giving [16], with donations to a charitable organization [24], and with the degree of providing verbal comfort and support (prosocial behavior) towards socially excluded individuals [25]. The association between empathy and prosocial behavior has been directly tested by either measuring empathy as an individual trait or, more directly, by inducing an empathic state, and estimating the associations of trait or state empathy to altruistic behavior [26]. The hypothesis has received empirical support, but the evidence is not unambiguous. A moderate positive association ($r = .38$) between empathy and prosocial behavior was estimated in a meta-analysis [27], but the estimate may be too high because of selective analysis and reporting common in social psychology [28].

Regarding *trait* empathy, empathic concern has been observed to be positively associated with volunteering to help others ($r = .31$) [29] and with offers in the public good game ($R^2 = 0.086$) [30]. In Jordan et al. [30], affective empathy was also positively associated with giving in a public good game but, curiously, *negatively* associated with giving to charity. A public good game is a social dilemma where individually rational behavior is in conflict with the pareto optimal strategy. For example, an individually rational behavior is to continue using fossil fuels, but refraining from their use would be collectively rational to mitigate climate change (for a review, see e.g. [31]; a meta-analysis on linear public good is reported in [32]).

Edele et al. [4] observed a positive association between affective empathy and empathic concern and allocations to a dictator game recipient ($r = .53$ and $\beta = .52$ for affective empathy; and $r = .35$ for empathic concern). They did not observe a statistically significant association between cognitive empathy and dictator game giving ($r = .31$ and $\beta = .24$). Other studies have not found statistically significant associations between trait empathy dimensions and dictator game allocations [33–35]. Overall, a rather robust finding seems to be that cognitive empathy cannot predict dictator game allocations, whereas affective empathy and empathic concern were linked to dictator game allocations in one study.

Results on the association between *state* empathy and altruistic sharing are also somewhat unclear. Oswald [29] observed that asking participants to pay attention to a target's feelings increased self-reported readiness to help a third person more than asking participants to pay attention to the target's thoughts (Cohen's $d = 1.30$) or to irrelevant details (Cohen's $d = 1.66$). Batson et al. [9] studied the effects of imagining one's *own* feelings in the other's position in comparison to imagining the *other's* feelings. They observed that the imagine-other task increased altruistic decisions.

Studies on the dictator game show that inducing empathy sometimes increases allocations to recipients. Klimecki et al. [36] observed that priming participants with videos increased

dictator game giving when the recipients were the suffering people in the videos ($R^2$ = .41). Saslow et al. [37] observed that compassion priming increased giving only among the less religious participants. Similarly, Powell et al. [34] showed images of suffering and vulnerability and found only subgroup effects on dictator game giving. A three-way interaction suggested that inducing compassion increased dictator game giving when trust was low and affective empathy was high. Lönnqvist and Walkowitz [38] elicited empathy with a writing task and observed that it increased reports of feeling empathetic, compassion, and concern, but there was no statistically significant effect on dictator game giving. The estimated effect size on giving was small ($r$ = .067) and the study reported statistical power above 99% to detect a typical social psychology effect size ($r$ = .20).

The overall conclusion about existing evidence on empathy and dictator game behavior is that neither trait nor state empathy predicts dictator game giving unequivocally but rather that the observed associations vary and may be conditional on the specific design of the experiment and dictators' personal characteristics.

## Hypotheses

We expect that r*ole awareness has an effect on dictator game giving; the donations will be greater under role uncertainty than under role certainty*. This expectation is based on evidence from earlier research as well as the assumption that role uncertainty will drive attention to both roles. Based on the empathy-altruism hypothesis we expect that *empathy induction will increase dictator game giving*. However, this expectation is somewhat tentative because existing evidence on dictator game giving does not give unequivocal support for it. We are interested in finding out whether there is an *interaction between role awareness and empathy induction*, indicating that the combination of these treatment conditions would be an effective way to increase dictator game giving. It is possible that together the two conditions produce a stronger stimulus for sharing compared to what either one of them does alone. We also expect that certain dimensions of trait empathy are positively associated to dictator game giving. Based on previous studies, it seems likely that affective empathy and empathic concern could increase sharing.

Finally, we are interested in exploring whether there is an interaction or a multiplicative effect between different trait empathy dimensions and the manipulations of role awareness and empathy induction. Regarding these interactions, there are processes that may work in opposite directions. On one hand, role awareness and state empathy manipulations may be effective especially among those participants who are sensitive to others' emotional states. On the other hand, people scoring high on empathy may not be influenced by these manipulations because they are inclined to share resources anyhow. Those who score low on trait empathy, in turn, are not likely to think about the feelings of others, in general, but may do so when role uncertainty or empathy induction prompts them to pay attention to others' position. There is some evidence which suggests that trait empathy may condition how people react to empathy or emotion induction. Eliciting emotions has been observed to affect dictator game giving in a different way depending on the trait empathy among children [39]. Another study suggests that empathy induction increases dictator game giving both among violent offenders and controls, but that lower self-reported trait empathy is associated to a smaller response to empathy induction [40].

It is furthermore possible that different dimensions of empathy interact differently with the treatments. There seems to be a reason to assume that the cognitive and affective dimensions of empathy are differently related to altruistic behavior. Cognitive empathy means an ability to understand others' emotions and predict their behavior, but it is not necessarily associated

with caring about others' welfare, whereas especially empathic concern or sympathy is conceptually linked to caring about others. We therefore ask *whether those who score high on a specific empathy dimension, say affective empathy, are in particular sensitive to the role awareness or empathy induction manipulation, or is it rather so that these types of people are likely to share money independent of the experimental manipulations.*

## Material and methods

### Participants

The sample consisted of 186 participants (127 female, 48 male, 11 other) who were recruited via an online system from a pool of registered subjects (ORSEE, [41]). We aimed at a similar sample size used in previous experimental studies measuring dictator game giving with real dictator-recipient pairs and a between subjects design [12], but the sample size was also influenced by our ability to recruit participants to the experiment within a single academic year. The members in the subject pool are mainly recruited through advertising on campus but people other than students are also able to sign in. Participants were mostly graduate or undergraduate students, (n = 167, mean age = 26.6 years, SD = 5.47, min = 19 years, max = 52 years), with various academic backgrounds (the arts, business, medical sciences, social sciences, and natural sciences). Twenty-one participants left the item 'subject' empty or listed occupation other than student in the background survey (mean age = 43.7 years, SD = 12.33, min = 21 years, max = 65 years).

### Measures

The invitation email included a link to an empathy questionnaire which consisted of items from two empathy measures, Davis's [42] Interpersonal Reactivity Index (IRI), and Reniers et al.'s [43] Questionnaire of Cognitive and Affective Empathy (QCAE). IRI is a widely used empathy measure, and QCAE separates the cognitive and affective aspects of empathy that we were interested in. As some items in QCAE are the same with IRI, they were asked only once. The Finnish translation of IRI has been tested before [44], whereas we are not aware of an existing Finnish translation of QCAE. We used a method where each item was first translated into Finnish and then back to English by another member of the research group to ensure the validity of the translation. IRI and QCAE items are listed in S1 File.

IRI includes four subscales, each with seven items: *perspective-taking* (cognitive empathy), *fantasy* (identification with fictional characters), *personal distress* (self-oriented feelings of anxiety and unease in tense interpersonal situations), and *empathic concern* (feeling for others or sympathy). QCAE contains two subscales which are further divided into components. Cognitive empathy subscale includes two components, *perspective-taking* (10 items), and *online simulation* (9 items), and *affective empathy* subscale three components: *emotion contagion*, *proximal responsivity*, and *peripheral responsivity*, each with 4 items.

### Dictator game

Within each experimental session, participants were randomly matched into dictator-recipient pairs as well as randomly assigned into the roles of a dictator or a recipient. Each dictator was endowed with 16 euros and was told that he or she could allocate the endowment freely between him or herself and a randomly matched recipient in one-euro increments. In total 186 subjects participated. Of these, 131 were assigned the role of a dictator. The independent and individual allocation decision made by dictators is the basic unit of our analysis (dictator game giving, n = 131). The rest of the subjects (n = 55) were recipients under the *role certainty*

condition. Under *role uncertainty*, all participants were instructed to make an allocation decision, and when all participants had made their decisions, a computer randomized the participants into the roles of decision-makers or recipients.

## Manipulations

For *empathy induction*, we used an imagination exercise similar to that of Batson et al. [9] and Lönnqvist and Walkowitz [38]. Participants were asked to imagine and write down what they thought the recipient would feel about different amounts of money the decision-maker would give to him or her.

Participants were instructed as follows:

*Before you decide how much you will send to the recipient, evaluate how the recipient will feel about receiving different sums of money. Write down your evaluation of the recipient's feelings.*

This instruction was followed by an open space where participants wrote their thoughts (maximum number of characters was restricted to 640). They could not proceed before writing something. A classification of the participants' written answers to the open question about the recipient's feelings is presented in S1 Table.

We selected an imagine-other task (e.g. imagine how low income people feel) rather than an imagine-self task (e.g. imagine how you would feel if you had low income) because there is evidence that the first is a more efficient method for inducing empathy and altruism [8]. The writing task was used to confirm that the participants really imagined and evaluated the recipient's feelings in their minds. In some previous studies, videos of suffering people have been used to elicit empathy [36, 37], but based on just face validity such stimuli are more likely to elicit sympathetic reactions such as pity, rather than experiencing and thinking about other people's feelings [45]. Lönnqvist and Walkowitz [38] provide evidence that empathy induction with a writing task increases reported feelings of empathy and sympathy.

Role awareness manipulation had two levels: role certainty and role uncertainty. Under *role certainty*, participants knew their roles when making their allocation decisions, and only dictators made allocation decisions. In contrast, under *role uncertainty*, participants were randomly assigned into two roles (dictator/recipient) only after each participant had made a dictator game allocation decision. In other words, each participant made an allocation decision but only dictators' decisions were implemented after roles were assigned. It is notable that in all four treatment cells, a payoff maximizing participant would allocate all money to him or herself. Moreover, under role uncertainty, risk attitudes should not be relevant predictors of choices.

## Design and procedures

The experiment required participation at two timepoints. Participants responded first to the empathy questionnaire online via a link provided to them when they signed up for the experiment. The questionnaire was filled out at least one week prior to the experimental session in which a participant took part. On average, the questionnaire was filled two weeks before the experiment. Only those participants who completed the empathy questionnaire were allowed to take part in the experiment.

The experiment comprised of 14 sessions, from September to November 2018, and in April 2019. Each participant took part in one experimental session only. Each session took place on

a single day, at the Decision-making Laboratory of the University of Turku, and lasted from 1h to 1.5h, including completion of the background survey and paying out the earnings.

The design was a 2×2 factorial experiment with *role awareness* (role certainty vs. role uncertainty) and *state empathy induction* (no empathy induction vs. empathy induction) as factors. This yielded four cells: *Baseline (Role Certainty/No Empathy Induction, RC/NEI)*, *Role Uncertainty (Role Uncertainty/No Empathy Induction, RU/NEI)*, *Empathy Induction (Role Certainty/ Empathy Induction, RC/EI)*, and *Role Uncertainty/Empathy Induction (RU/EI)* (Table 1). The number of observations per cell is slightly unbalanced because under role uncertainty all participants made choices, whereas under role certainty only half of the participants, i.e. those assigned to a dictator role with certainty, made choices.

Upon arrival in the laboratory, each participant was randomly assigned to a cubicle and general instructions were read aloud before starting the experimental session. An English translation of the instructions is presented in S2 File. Under role certainty, randomization to the roles was done before participants made their choices, whereas under role uncertainty, it was done only after the allocation decisions were made. The experiment was anonymous and computerized (with Z-tree, [46]).

In each experimental session, participants were randomly and anonymously paired into groups of two by the computer. Under role certainty, each participant in each pair was instructed of his or her role (dictator or recipient) before the actual dictator game was played out. Dictators thereby knew that their allocation decisions would be realized with certainty. In contrast, under role uncertainty, each participant was instructed that he or she would be randomly assigned to either role (dictator or recipient), and that the realized role would be revealed only after he or she had made the allocation decision. This meant that decision-makers knew that their allocation decisions may not be implemented, and that allocation would instead be made according to the decision of another person. Under empathy induction, participants were asked to write down their thoughts about the feelings of the recipient in an open space provided on the computer screen.

After having played the dictator game once, participants completed a questionnaire including an argument evaluation task not related to the experiment, standard measures of trust, political interest, social wellbeing, and attachment style [47], as well as background variables. The questionnaire consisted of a total of 70 items and is available from the authors. As the questionnaire was part of a separate unrelated study, we have not explored any of the possible associations between the collected survey data and experimental results apart from using gender as a control variable in regression analysis. After the completion of the questionnaire, participants were paid what they had earned from the dictator game and a show-up fee of four euros.

The research was conducted according to the ethical principles of the Finnish Advisory Board for Research Integrity (http://www.tenk.fi/). According to Finnish regulations, specific ethics approval was not necessary for this study. An informed consent was obtained from all participants: the recruitment web page includes a register description and written rules of

**Table 1. Experimental design.**

| | Role certainty (n = 55) | Role uncertainty (n = 76) |
|---|---|---|
| No empathy induction (n = 64) | *Baseline* | *RU/NEI* |
| | *RC/NEI* (regular DG) | n = 36 dictators |
| | n = 28 dictators | |
| Empathy induction (n = 67) | *RC/EI* | *RU/EI* |
| | n = 27 dictators | n = 40 dictators |

participation which describe e.g. how data will be managed and how anonymity will be preserved (http://pcrclab.utu.fi/public/index.php). It is made clear that the participants indicate their consent to these rules upon subscribing to the register. All participants were adults.

## Statistical power

Statistically insignificant results can be the outcome of low statistical power to observe relevant effect sizes. Based on the study design and the realized sample size, we ran power analyses for various effect sizes, independent of our observed effect size estimate. These are therefore not post-hoc power estimates which are dependent on the observed effect size. Assuming our study design and sample size, we calculated statistical powers to observe conventionally small (Cohen's $d = 0.20$), moderate ($d = 0.50$), and large ($d = 0.80$) effect sizes as well as one very large reported in the literature ($d = 1.3$; [29]). For the main effect of empathy induction, our design had powers of 20% (small effect size), 81% (moderate), 99% (large), and 100% (very large). For role awareness, the powers were 20% (small), 80% (moderate), 99% (large), and 100% (very large). For the interaction of empathy induction and role awareness, the powers were 9% (small), 30% (moderate), 60% (large), and 95% (very large).

## Results

### Dictators' allocation behavior

Across all treatment conditions, the dictators allocated on average 42% or 6.70 EUR of their initial endowment. In particular, the dictators sent *at least* half of their initial endowment in 58% of the cases (in 76 out of 131 allocation decisions). The modal amount allocated was 8 EUR in each cell, i.e. a half of the 16 EUR endowment. Table 2 represents the average contributions per treatment and per treatment cell as well as average scores of the empathy indexes.

Allocation behavior was analyzed with a 2 (role certainty vs. role uncertainty) × 2 (no empathy induction vs. empathy induction) ANOVA. The results revealed that the main effect of the

**Table 2. Descriptive statistics of employed measures.**

|  | M | Md | Min | Max | SD |
|---|---|---|---|---|---|
| **EUR allocated per factor level (main effects)** |  |  |  |  |  |
| RC | 5.47 | 6 | 0 | 9 | 2.95 |
| RU | 7.61 | 8 | 0 | 16 | 2.81 |
| NEI | 6.19 | 8 | 0 | 11 | 3.00 |
| EI | 7.21 | 8 | 0 | 16 | 3.03 |
| **EUR allocated in treatment cells** |  |  |  |  |  |
| Baseline, *RC/NEI* | 5.36 | 6.5 | 0 | 9 | 3.11 |
| *RC/EI* | 5.59 | 6 | 0 | 81 | 2.83 |
| *RU/NEI* | 6.83 | 8 | 0 | 11 | 2.79 |
| *RU/EI* | 8.30 | 8 | 4 | 16 | 2.68 |
| **IRI scores** |  |  |  |  |  |
| Perspective taking | 17.86 | 18 | 7 | 27 | 4.05 |
| Fantasy | 17.95 | 18 | 5 | 28 | 5.35 |
| Personal distress | 12.66 | 13 | 1 | 28 | 4.58 |
| Empathetic concern | 19.30 | 20 | 7 | 28 | 4.29 |
| **QCAE scores** |  |  |  |  |  |
| Cognitive empathy | 49.54 | 50 | 14 | 69 | 9.92 |
| Affective empathy | 31.58 | 32 | 18 | 44 | 6.17 |

**Table 3. A two-way analysis of variance (type II) on the treatment effects.**

| Treatment Effect | D.F. | Sum of Sq. | F-value | Pr(>F) |
|---|---|---|---|---|
| **Role Awareness** | 1 | 140.4 | 17.44 | 5.47E-05 |
| **Empathy Induction** | 1 | 29.4 | 3.66 | 0.058 |
| **Role Awareness x Empathy Induction** | 1 | 12.1 | 1.50 | 0.223 |
| **Residuals** | 127 | 1022.3 | | |

role awareness treatment was highly significant, F(1, 127) = 17.44, *p* = 0.000055 (type II sum of squares, additive effect). Under role certainty, the mean dictator allocation was 5.47 (SD = 2.95) EUR, whereas under role uncertainty, it was 7.61 (SD = 2.81) EUR. The effect of the empathy induction manipulation, F(1, 127) = 3.66, *p* = 0.058, was not statistically significant. In the no empathy induction condition, dictators' mean allocation was 6.19 (SD = 3.00) EUR, whereas empathy primed dictators' mean allocation was 7.21 EUR (SD = 3.03) (cf. Table 2). The interaction effect between role uncertainty and empathy induction was not significant (F(1, 127) = 1.50 *p* = 0.223) (see Table 3 reporting the results of the two way ANOVA estimation).

The residuals passed a homogenous variance test (Levene's Test, F(3,127) = 1.6, p = .19) but were not exactly normal (Shapiro-Wilk, W = 0.95, p = .0001).

**Distribution of allocations.**  Under role certainty (RC), 40% (22 out of 55) of dictators sent 8 EUR to their recipient, and one dictator allocated more than that, that is, 42% of the participants sent at least half of their endowment in this treatment. Under role uncertainty (RU), sharing half (n = 40) or more than half (n = 13) of the endowment took place in 70% (53 out of 76) of the cases. The difference in observed proportions of subjects sharing at least half of the endowment (42% vs. 70%) is statistically significant ($X^2$ = 9.10, *df* = 1, *p* = 0.0026). The respective difference in proportions between no empathy induction (NEI) (52%) and empathy induction (EI) (62%) treatments was not significant ($X^2$ = 1.65, *df* = 1, *p* = 0.199). The distribution of the allocation decisions in each cell is depicted in Fig 1 (cf. Table 2).

We were interested in the difference between the sizes of the effects of empathy induction and role awareness on giving. It is possible that there is no statistically significant difference in the effect sizes of the two manipulations even when we have the result that role awareness has a statistically significant effect whereas empathy induction does not. First, for individual manipulations, a lack of statistically significant effect is not yet evidence against the effect in the Fisherian or NHST type of inference [48]. When a non-significant effect occurs, there are no statistical guarantees or error limits for non-zero effects in the population. Second, we cannot make statistical inferences about the differences between the sizes of two coefficients directly from their individual statistical significance in a regression equation. We need another test for that purpose.

We used a *t*-test and equivalence testing (Two One-Sided T-tests, TOST; [49]) to answer the questions whether the difference in the effect sizes of empathy induction and role awareness is different from zero and whether the difference is smaller than a moderate effect size (Cohen's *d* = 0.5). If the difference is smaller than moderate, we can consider the two methods to be equivalent in their effects. The contrast or difference in the effects [(*RU/NEI–RC/NEI*)–(*RC/EI–RC/NEI*)] reduces to the difference *RU/NEI–RC/EI* which can be tested with a *t*-test. The difference from zero or the null hypothesis test was non-significant, *t*(55.72) = 1.732, *p* = 0.089. The equivalence test was also non-significant, *t*(55.72) = -0.229, *p* = .410, given equivalence bounds of -1.405 and 1.405. Based on the equivalence test and the null-hypothesis test combined, we can conclude that the observed effect is not statistically different from zero

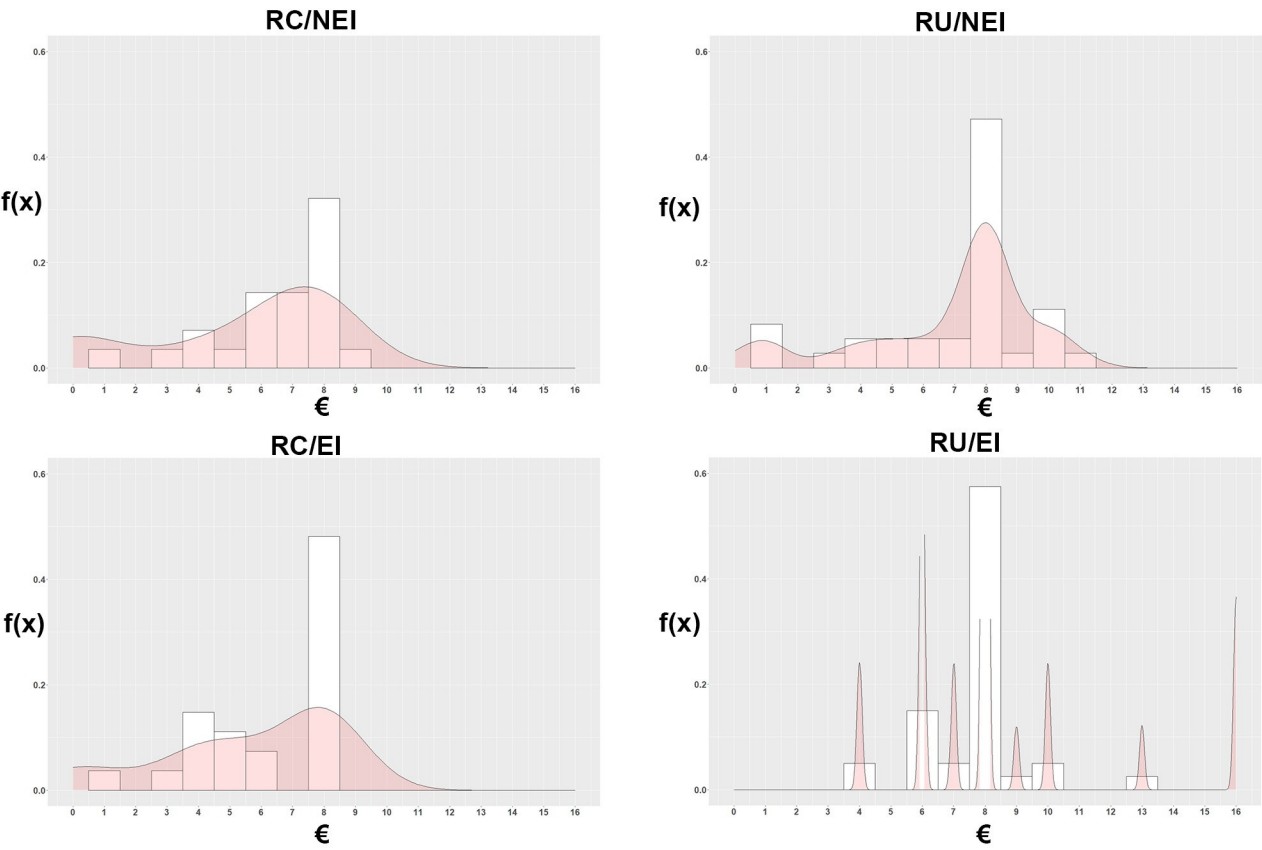

**Fig 1. Distribution of allocations in each cell.**

and not statistically equivalent to zero. That is, we do not have evidence that the effects are the same nor that the effects are different. More precise measurements or larger sample sizes are likely to help in deciding whether the effects of role awareness and empathy are equivalent.

### Trait empathy measures and dictator game allocations

In order to investigate whether trait empathy modulates dictator game giving, we first computed the raw Spearman correlation coefficients between the scores on the four subscales of IRI and dictator game giving as well as between the scores on the two empathy subscales of QCAE and dictator game giving. This was done independent of the participants' treatment conditions. None of these measures were statistically significantly related to dictators' allocations to the recipients, as shown in Table 4.

In order to further explore how the empathy measures were related to sharing in the dictator game, we ran automated stepwise regressions (method "Backwards" with two degrees of freedom used for the penalty) with dictator game giving as the dependent variable.

In the first set of stepwise regressions, IRI subscales Concern and Perspective taking, treatment variable dummies, and their interaction terms were included as (unstandardized) predictors. Personal Distress and Fantasy scales of IRI were not included in the models because we were interested in affective and cognitive empathy. There is psychometric evidence against combining all the four scales into a two factor model of affective and cognitive empathy [50]. Empathic Concern and Perspective Taking dimensions measure affective and cognitive

**Table 4. Spearman correlation coefficients between allocation decision and empathy measures.**

| Empathy subscale | $\rho^a$ | p-val. |
|---|---|---|
| IRI | | |
| Perspective | 0.12 | 0.13 |
| Fantasy | 0.11 | 0.17 |
| Distress | 0.04 | 0.65 |
| Concern | 0.14 | 0.10 |
| QCAE | | |
| Cognitive | 0.07 | 0.50 |
| Affective | 0.11 | 0.25 |

$^a\rho$ = Spearman's *rho* correlation coefficient.

empathy best out of the four IRI scales. In the second set of regressions, the two QCAE empathy subscales (Cognitive and Affective), treatment variable dummies and their interaction terms functioned as (unstandardized) predictors, respectively. In both sets of models, gender was included as a simple control variable. There were only the two-way interactions in the starting point models (no higher order interaction terms were included), and the empathy variables did not have interaction terms between themselves.

In the first set of regressions, IRI Concern subscale and the treatment variable dummies survived the iterated elimination based on Akaike's Information Criterion. We used the step function in R's stats (version 3.6.2) package, that computes Akaike's Information Criterion for one or several fitted model objects [51]. This was also true for the interaction terms between Concern and the treatment conditions. However, only the coefficients of the role awareness treatment dummy and its interaction term had significant exploratory *p*-values in the final model. The starting point of the full model (I) and the final model (II) with just the remaining treatment manipulation dummy variables are reported in Table 5.

Fig 2 illustrates the interaction between empathic concern and role awareness. Under role uncertainty, the amount of money allocated to the recipient decreases slightly as empathic concern increases. Under role certainty, the amount of money allocated to the recipient increases as empathic concern increases. In the range of the IRI empathic concern scale [0, 28], role uncertainty increased the amount of money given only for the participants with relatively low scores.

In the second set of models where the QCAE subscales were assigned as predictors, the affective empathy score [0, 48] remained a statistically significant predictor ($t = 2.75$, $p = 0.007$) after the variables whose inclusion did not improve the model's explanatory power were eliminated. This was also the case for the interaction term between affective empathy and role awareness ($t = -2.66$, $p = 0.009$). Moreover, both treatment manipulation dummies survived the process of elimination, and are thus included in the final regression specification, although the coefficient of the empathy induction dummy was not significant ($t = 1.70$, $p = 0.092$). The final model and its starting point are shown in Table 6, models III and IV.

Fig 3 illustrates the relationship between QCAE's affective empathy subscale and dictator game behavior. Under role certainty, for each additional point in affective empathy score, the amount of money allocated to the recipient increases 15 cents (model IV), as shown by the red line (Fig 3). However, under the role uncertainty (cyan line), we observe an opposite effect: as the affective empathy score increases the allocations decrease. Within the measurement interval, the effect of role awareness is about zero when affective empathy is the highest (right hand

**Table 5. Regression results of dictator game giving, IRI's subscales as predictors.**

| Model I | Estimate | S.E. | 95% CI [LL; UL] | Pr(>\|t\|) | |
|---|---|---|---|---|---|
| Intercept | -1.288 | 6.317 | [-13.83, 11.25] | 0.839 | |
| Concern | 0.38 | 0.256 | [-0.13, 0.89] | 0.141 | |
| Perspective | -0.129 | 0.312 | [-0.75, 0.49] | 0.681 | |
| Role awareness | 7.696 | 3.591 | [0.57, 14.83] | 0.035 | |
| Empathy induction | -4.486 | 3.206 | [-10.85, 1.88] | 0.165 | |
| Con. x Role Awar. | -0.352 | 0.153 | [-0.66, -0.05] | 0.024 | |
| Con. x Emp. Ind. | 0.145 | 0.152 | [-0.16, 0.45] | 0.343 | |
| Pers. x Role Awar. | 0.003 | 0.164 | [-0.32, 0.32] | 0.987 | |
| Pers. x Emp. Ind. | 0.078 | 0.159 | [-0.24, 0.39] | 0.624 | |
| Role Awar. x Emp. Ind. | 0.743 | 1.134 | [-1.51, 3.00] | 0.514 | |
| Gender | 0.241 | 0.559 | [-0.86, 1.35] | 0.667 | |
| $R^2$ adj. 0.186 | | | | | |
| F = 3.37, DF (10 and 94) | | | | | |
| *p*-val. = 0.00087 | | | | | |
| Model II (final) | Estimate | S.E. | 95% CI [LL; UL] | Pr(>\|t\|) | Cohen's *f2* |
| Intercept | 2.172 | 1.809 | [-1.41, 5.76] | 0.233 | 0.014 |
| Concern | 0.154 | 0.093 | [-0.03, 0.34] | 0.10 | 0.027 |
| Role Awareness | 9.192 | 2.475 | [4.28, 14.10] | 0.0003 | 0.14 |
| Empathy Induction | 2.915 | 2.439 | [-7.75, 1.92] | 0.235 | 0.014 |
| Con. x Role Awar. | 0.363 | 0.124 | [-0.61, -0.12] | 0.004 | 0.084 |
| Con. x Emp. Ind. | 0.191 | 0.122 | [-0.05, 0.43] | 0.122 | 0.024 |
| $R^2$ adj. 0.2203 | | | | | |
| F = 6.88, DF (5 and 99) | | | | | |
| *p*-val. = 0.000015 | | | | | |

side of the picture) and the effect of role awareness increases as affective empathy decreases. The effect of empathy induction is not statistically significant.

## Discussion

We tested the direct effects of role awareness, state empathy and trait empathy on dictator game giving as well as the interaction effect between role awareness and state empathy manipulations and different dimensions of trait empathy. We replicated Iriberri and Rey-Biel's [12] results which show that role uncertainty increases allocations to the dictator game recipient. As in other recent studies [34, 37, 38], induction of state empathy did not have a statistically significant main effect on dictator game giving. Furthermore, the interaction between role awareness and empathy induction was not statistically significant. Apart from affective empathy, IRI and QCAE subscales did not have direct effects on dictator game giving. A lack of statistically significant associations between dictator game allocations and trait empathy have also been reported in other recent studies [33–35]. However, we observed an interaction effect between role awareness and empathy subscales: dictator game donations were greater under *role uncertainty than under role certainty when trait empathic concern or trait affective empathy was low*.

Based on our power calculations we can be confident in concluding that there are no large main effects of empathy induction nor very large interaction effects between empathy induction and role awareness. Our effect size estimate of the empathy induction was smallish (Cohen's *d* = 0.34), and we should not think that the study provides evidence against the

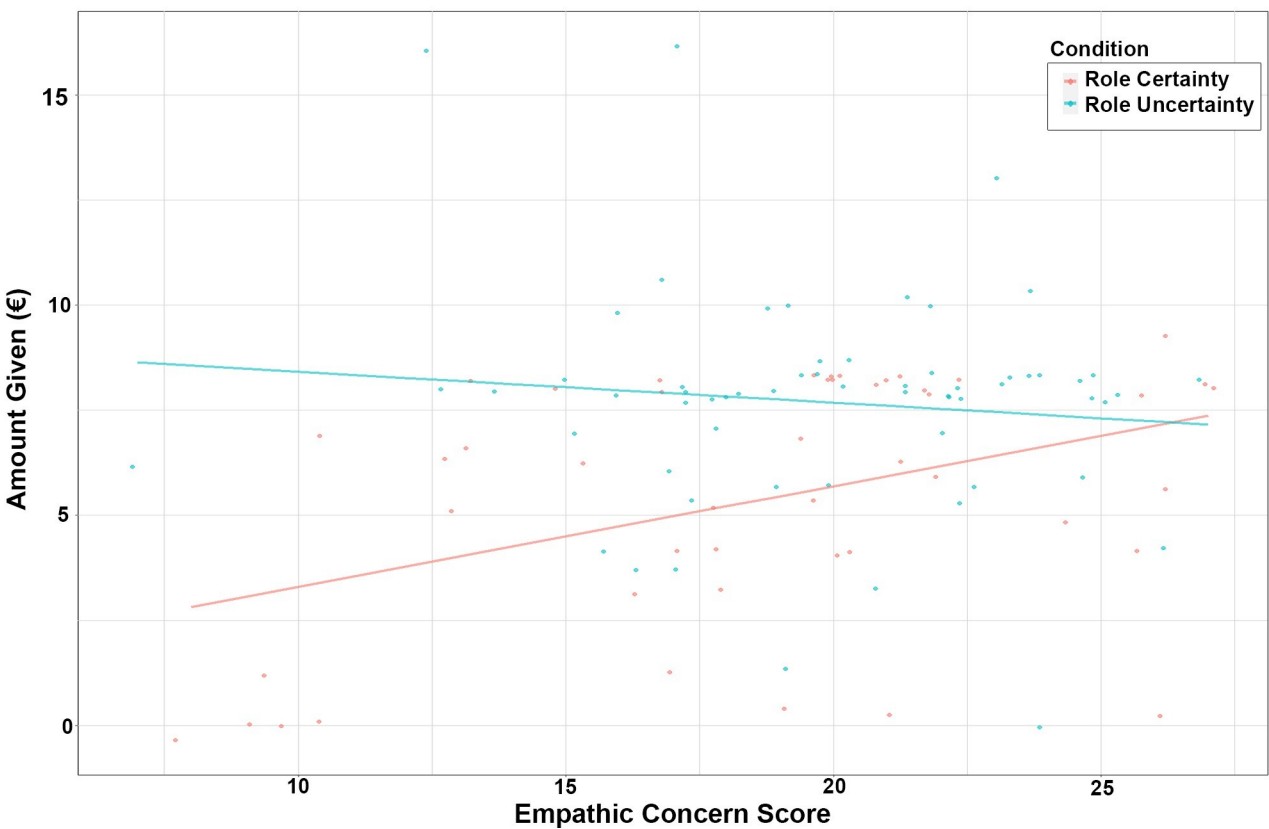

**Fig 2. Dictator game giving predicted by the empathic concern score (cyan line: Role uncertainty, red line: Role certainty).**

presence of small effects of empathy induction or its interaction with role awareness. We come to a similar conclusion as Lönnqvist and Walkowitz [38] that the size of the effect of empathy induction on dictator game giving is likely to be smaller than moderate ($d = 0.50$) and certainly less than very large ($d = 1.3$).

Admittedly, our operationalization of empathy induction was not very strong. We used the same empathy induction method as Lönnqvist and Walkowitz [38], i.e., imaging and writing down the recipients' feelings, and had the same statistically non-significant association to dictator game giving. Our empathy induction is cognitive in nature because it asks decision-makers to imagine how the target feels, and it does not directly induce experienced feeling for the target. While the induction may induce such feelings among some of the decision-makers, it may primarily fail to do so. It is therefore relevant to consider whether an affective empathy manipulation would have been more efficient in generating dictator game giving. In some dictator game studies, empathy has been induced by showing suffering persons in need. However, these types of strong visual stimuli have not produced statistically significant main effects either [34, 37], with the exception of Klimecki et al. [36] where the participants may have thought that they were giving their money to the suffering people appearing in the videos rather than anonymous persons in the laboratory, as in Saslow et al. [37] or Powell et al. [34]. It is possible that, rather than telling us about empathy, or sympathy, the statistically significant results of Klimecki et al. [36] are the effect of a norm of helping suffering people, a norm that does not apply to people without signals of suffering. Overall, these results suggest that, if it exists, the effect of empathy induction may be small and requires a large sample size to be

**Table 6. Regression results of dictator game giving, QCAE subscales as predictors.**

| Model III | Estimate | S.E. | 95% CI [LL; UL] | Pr(>|t|) | |
|---|---|---|---|---|---|
| Intercept | -4.15 | 7.37 | [-18.78, 10.49] | 0.57 | |
| Role Awareness | 7.92 | 3.94 | [0.11, 15.74] | 0.05 | |
| Empathy Induction | -2.93 | 3.58 | [-10.03, 0.85] | 0.42 | |
| Cognitive Empathy | -0.11 | 0.15 | [-0.42, 0.19] | 0.46 | |
| Affective Empathy | 0.41 | 0.22 | [-0.02, 0.85] | 0.06 | |
| Cog. Emp. x Role Awar. | 0.002 | 0.08 | [-0.15, 0.15] | 0.98 | |
| Cog. Emp. x Emp. Ind. | 0.08 | 0.07 | [-0.06, 0.21] | 0.26 | |
| Aff. Emp. x Role Awar. | -0.23 | 0.11 | [-0.44, -0.004] | 0.05 | |
| Aff. Emp. x Emp. Ind. | -0.04 | 0.11 | [-0.25, 0.18] | 0.74 | |
| Role Awar.. x Emp. Ind. | 0.74 | 1.16 | [-1.56, 3.03] | 0.52 | |
| Gender | 0.41 | 0.61 | [-0.81, 1.62] | 0.51 | |
| $R^2$ (adj.) 0.159 | | | | | |
| F = 2.97, d.f. (10, 94) | | | | | |
| *p* = 0.00274 | | | | | |
| **Model IV (final)** | **Estimate** | **S.E.** | **95% CI [LL; UL]** | **Pr(>|t|)** | **Cohen's f2** |
| Intercept | 0.37 | 1.96 | [-3.51, 4.25] | 0.85 | 0.0035 |
| Role Awareness | 9.33 | 2.68 | [4.02, 14.64] | 0.0011 | 0.12 |
| Empathy Induction | 0.85 | 0.53 | [-0.20, 1.89] | 0.092 | 0.025 |
| Affective Empathy | 0.38 | 0.14 | [0.028, 0.27] | 0.007 | 0.057 |
| Affective Empathy x Role Awareness | -0.23 | 0.08 | [-0.40, -0.06] | 0.009 | 0.07 |
| $R^2$ (adj.) 0.191 | | | | | |
| F = 7.13, d.f. (4, 103) | | | | | |
| *p* = 0.000043 | | | | | |

detected as statistically significant. A comparison of an affective and cognitive empathy induction would still be worthwhile in future studies.

It is also noteworthy that our empathy induction may have had counterproductive consequences if respondents felt that the writing task gave them an opportunity to *explain* their self-interested behavior to the experimenter. There is evidence that people are motivated by a desire to be seen as fair and that this desire influences dictator game giving [52]. Instead of driving attention to the recipient's feelings, our empathy induction may have led some dictators to give explanations for not being fair.

In our experiment, adding empathy induction to role uncertainty did not boost altruistic sharing over what role uncertainty did alone. However, this result should be interpreted with some caution because of the possible ceiling effect, and the 20% power to have a statistical significance for small effect sizes reported in the literature. In possible future studies, having larger monetary sums could reduce the ceiling effects and larger sample sizes are needed for better statistical power.

Why did role awareness influence dictator game giving? When role uncertainty is applied, each role can potentially realize which possibly induces perspective-taking, i.e. picturing oneself in the position of the recipient. This type of perspective-taking may increase allocations to the recipient [27]. Another possible route to increased dictator giving game may be an attenuated experience of being in a position of power. When both roles can be realized with equal probability, power over the decision is uncertain. There is some evidence that being in a powerful position decreases the tendency to see things from another's perspective [53–55], and if this is true, reducing the feeling of power could potentially increase perspective-taking, and via

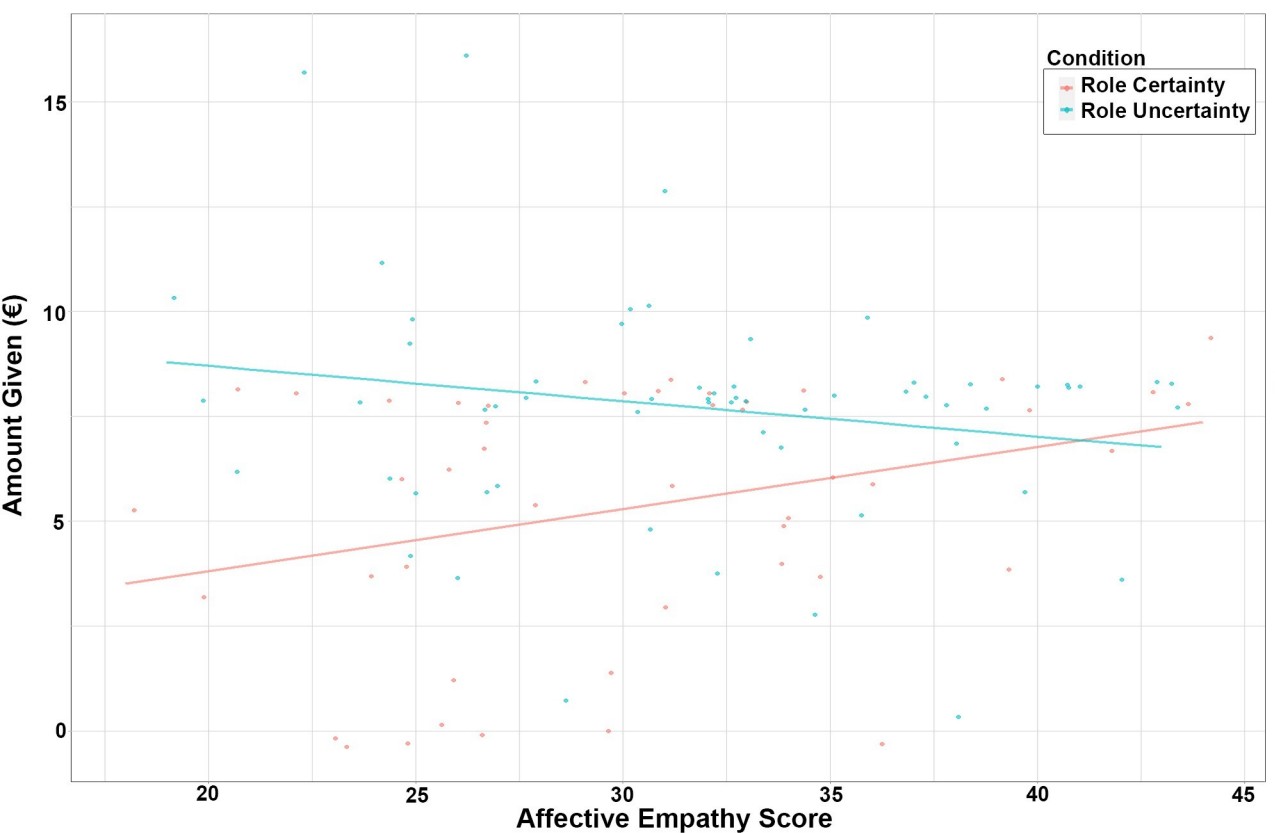

**Fig 3. Dictator game giving predicted by the affective empathy score (cyan line: Role uncertainty, red line: Role certainty).**

that process, dictator game giving. In effect, Mesa-Vazquez et al. [56] observed a decrease in dictator game giving when the probability of being the dictator increased.

Regarding the effects of trait empathy on dictator game allocations, our study suggests that affective empathy is associated with more dictator game giving under the role certainty condition, usually used in dictator games. We therefore replicate Edele et al.'s [4] observation. They measured affective empathy with IRI's empathic concern scale and a photo-based empathy test. Further, like us, Edele et al. did not observe a statistically significant connection between IRI's perspective taking scale and dictator game giving. However, it is noteworthy that if the true effect sizes of the trait measures are small, our statistical power to detect them was low.

We observed an interaction between role awareness and QCAE's trait affective empathy as well as IRI's empathic concern. Within the scale range, role awareness had a smaller influence on dictator game behavior for those participants who had higher levels of affective empathy or empathic concern. This result suggests that those with higher self-reported affective empathy or empathic concern tend to see things from others' perspective and to be emotionally affected by these perspectives even without specific methods to stimulate perspective-taking. Those with lower affective empathy or empathic concern, in turn, tend to consider choice tasks from their own perspective alone, but role uncertainty may direct their attention to the recipient's position, and via that process, increase allocations to the recipient. This proposal is compatible with views emphasizing that prosocial behavior is motivated by many separate variables and their interdependent relations [57]. Previous studies have observed similar interaction effects with personality types and empathy induction. The results of Powell et al. [34] suggest that

inducing compassion increases dictator game giving when trust is low and affective empathy is high, whereas Saslow et al. [37] observed that compassion priming increased giving in a hypothetical dictator game only among the less religious participants. Powell et al. conclude that inducing specific economic decisions, such as dictator game giving, may be most effective when it is tailored, taking both contextual factors and individual differences into account. The observation that the effect of role awareness is conditional on individuals' trait empathy levels supports the interpretation that role uncertainty works via perspective-taking: if this is the case, those who tend to be concerned by others' well-being anyhow are not much influenced by role awareness, whereas those who are not so likely to care about others tend to do so when uncertainty about their own role directs attention to both positions.

Regarding the practical implications of our results, it is noteworthy that manipulations of both empathy and role awareness can be regarded as methods for perspective-taking. These methods aim at widening individuals' perspectives from an in-group, or private, perspective to understanding of an out-group's point of view [8], and they can be deliberately used, e.g. in political rhetoric or by charities asking for donations, to increase the desired behavior. However, manipulations of role awareness and empathy are also different, because unlike empathy induction, role uncertainty is harder to implement outside the laboratory. In real life conditions, role uncertainty can only be approximated by using uncertainty about one's own position as a rhetorical instrument: "Think about how you would decide if you did not know your own role". Perspective-taking methods can be regarded mainly cognitive in nature, and since in our case role uncertainty produced an effect and empathy induction did not, it seems that the distinction between cognitive and affective methods is not alone relevant. Like cognitive trait measures, our empathy induction did not produce large effects [38]. As said, it is possible that dictators used the writing space to explain their self-interested behavior. The development of precise and powerful affective empathy induction methods would be useful for providing more conclusive evidence of the presence, or absence [34, 37], of a causal effect of affective empathy on altruistic behavior. However, it is noteworthy that evidence on the explanatory power of empathy subscales on different types of outcome variables is not extensive, and further studies are needed to get more robust results.

## Supporting information

**S1 Table. Classification of answers in the imagine other task by treatment.**
(DOCX)

**S1 File. Trait empathy scales.**
(DOCX)

**S2 File. On screen instructions.**
(DOCX)

## Acknowledgments

We would like to thank PLOS ONE's anonymous referees and the participants of workshops and conferences who have provided valuable comments on this paper. We also thank Laura Mattinen and Sanni Suikkanen for assistance in running the experimental sessions.

## Author Contributions

**Conceptualization:** Kaisa Herne, Jari K. Hietanen, Olli Lappalainen, Esa Palosaari.

**Data curation:** Olli Lappalainen, Esa Palosaari.

**Formal analysis:** Olli Lappalainen, Esa Palosaari.

**Funding acquisition:** Kaisa Herne.

**Investigation:** Olli Lappalainen.

**Methodology:** Kaisa Herne, Olli Lappalainen, Esa Palosaari.

**Project administration:** Kaisa Herne.

**Software:** Olli Lappalainen.

**Visualization:** Olli Lappalainen.

**Writing – original draft:** Kaisa Herne, Jari K. Hietanen, Olli Lappalainen, Esa Palosaari.

**Writing – review & editing:** Kaisa Herne, Jari K. Hietanen, Olli Lappalainen, Esa Palosaari.

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
