## [Decision Letter · Decision Letter 0]

28 Jul 2021

PONE-D-21-15135

The influence of role uncertainty, empathy induction and trait empathy on dictator game giving

PLOS ONE

Dear Dr. Herne,

Thank you for submitting your manuscript to PLOS ONE. After careful consideration, we feel that it has merit but does not fully meet PLOS ONE’s publication criteria as it currently stands. Therefore, we invite you to submit a revised version of the manuscript that addresses the points raised during the review process.

Please address all comments from the reviewers and revise the manuscript accordingly.

We look forward to receiving your revised manuscript.

Kind regards,

Marco Iacoboni

Academic Editor

PLOS ONE

Additional Editor Comments (if provided):

Reviewers' comments:

Reviewer's Responses to Questions

**Comments to the Author**

1. Is the manuscript technically sound, and do the data support the conclusions?

Reviewer #1: Partly

Reviewer #2: Partly

2. Has the statistical analysis been performed appropriately and rigorously? 

Reviewer #1: I Don't Know

Reviewer #2: Yes

3. Have the authors made all data underlying the findings in their manuscript fully available?

Reviewer #1: No

Reviewer #2: Yes

4. Is the manuscript presented in an intelligible fashion and written in standard English?

Reviewer #1: Yes

Reviewer #2: Yes

5. Review Comments to the Author

Reviewer #1: Thank you for inviting me to review “The influence of role uncertainty, empathy induction, and trait empathy on dictator game giving.” Herne et al. take a somewhat exploratory approach to address an interesting question: how state empathy, trait empathy, and role uncertainty influence dictator game giving. The researchers manipulated two factors: role uncertainty (information about one’s role) and empathy induction (written responses as to recipient’s feelings). These factors are well-motivated despite the sparse and discrepant findings in the literature. Additionally, the researchers extended the design and measured how participant-levels of empathy, state and trait, interact with the above factors. The research question is of open interest, the methods appear sound, and the results replicate prior work as well as introduce some novel findings. However, the following points require clarification before publication.

Abstract:

Line 27: “Each participant was randomly assigned into one of four conditions and into one of two roles, before or after decisions were made.” Are there not only four conditions, that include the role type? Please clarify this sentence

Introduction:

- Line 183: The researchers state their “main interest lies in detecting an interaction effect between different trait empathy dimensions and manipulations of role uncertainty and empathy induction.” However, the interaction focus is not clearly motivated in the introduction. It reads secondary to the main effects and, in fact, the researchers actually predict an additive role of the factors (lines 175 -177). Please provide more evidence for hypothesizing the interaction, and motivate the rationale earlier in the introduction, if the interaction is indeed the main interest of the study.

- It would be helpful to adopt a slightly broader motivation in the introduction, particularly to motivate the DV. Rather than stating the purpose is to “increase dictator game giving”, it would be helpful to describe what this means. Is dictator game giving an index of prosociality, i.e., at the cost to oneself? Or altruism? A latent operationalization is needed.

- The hypotheses (under “Research Questions”) read a little disorganized and verbose. Please streamline.

Methods:

- Line 205: include standard deviation

- Line 249: an example in this format- (e.g., XXXX) - for imagine-other and imagine-self tasks would be helpful.

- Please justify the unbalanced design. Why are there more participants in the “uncertain role” manipulation, and specifically, with the empathy induction condition?

Results

- Please specify the total participants included in the analysis, especially when considering all the individual difference measures.

- Line 341: I don’t understand the motivation behind including an effect size difference test, when one main effect is not significant (empathy induction). The rationale of the test should be more clearly motivated, rather than explaining what the analysis does (i.e., significance test from zero, line 343) or that the “effects are the same nor that the effects are different” (line 352).

- Line 368: please specify whether the predictors were standardized or unstandardized

- Line 368: how were the initial predictors in the stepwise regression selected?

- Please add confidence intervals and effect sizes

Discussion:

- Line 421 – 423: “role uncertainty was effective only when”- effective on what? Please be explicit.

- Please rephrase the paragraph starting at line 483 to avoid confusability.

- Line 484 does not follow clearly. Please correct to: “within the scale range, role uncertainty had a smaller influence on DG behavior (i.e., your DV) for those…”)

- Lines 485- 492: Interpretation of interaction effect is somewhat speculative, although logical. I would suggest that the authors include some relevant citations.

- Line 469: please add citation after recipient.

Please tone down the language in certain conclusions. For instance, line 497: “those who tend to be concerned by others are not influenced by role uncertainty…”

Line 510: The researchers state: “our results suggest that affective rather than cognitive processes enhance altruistic behavior.” Using the Dictator game as an index for altruistic behavior, is never clearly described nor is the focus on altruism clearly motivated in the introduction. Altruistic behavior should be operationalized and the researchers should explain how and why the dictator game measures this much earlier in the manuscript.

Minor:

Please improve the readability of the manuscript. Some minor suggestions are listed below.

- Line 43: change “charities is” to “charities are”

- Line 79: Get rid of question mark

- Line 83: change “as” to “or”

- Line 123: “public good game”. What is this? Helpful to include an example (e.g., XXXX)

Reviewer #2: The authors examined the roles of and interactions between empathy induction and role uncertainty on sharing behavior in an anonymized, one-shot dictator game, and examined the mediating role of trait empathy on these main effects using an across-subject design. By and large this study is well-designed, well-powered, and the grounding theoretical hypothesis is compelling: that role uncertainty promotes a form of perspective-taking (which could be argued to be a more effective, implicit form of empathy induction).

Regarding the empathy induction: While the authors do provide supporting evidence for the format of the empathy induction, they should address the seemingly cognitive nature of the induction, in light of the greater influence of affective empathy on empathic concern (as mentioned in the introduction). Future studies on this subject should perhaps include a more somatomotor and affective empathy induction protocol.

Regarding the statistical analyses: Did the authors test for normality of offers? Depending on this factor, they may have been better off reporting the median primarily and using non-parametric tests. Simply assuming normality can lead to erroneous conclusions.

Also, while the mean and median may be non-significantly different between some of the conditions, there may be significant differences between the shape of the distributions, and vice-versa. I recommend running a Kolmogorov–Smirnov test to ascertain whether this is the case.

Regarding the background/introduction, the authors present evidence that cognitive and affective empathy are subserved by distinct systems. While lesion and activation studies do support this interpretation, the issue may be more complex than presented: the neural bases of affective and cognitive empathy processes interact considerably. Recent research suggests that somatomotor and affective processing contribute to our evaluations of others’ internal states, beliefs, and intentions (Gallese, 2007; Schulte-Rüther et al., 2007; Frith and Singer, 2008; Obhi, 2012; Christov-Moore and Iacoboni, 2016; Christov-Moore et al., 2017a), as well as our decisions about others’ welfare (Greene, 2001; Camerer, 2003; Van’t Wout et al., 2006; Oullier and Basso, 2010; Hewig et al., 2011; Christov- Moore et al., 2017). Conversely, cognitive processes are increasingly implicated in the contextual modulation of neural resonance, a putative substrate of affective empathy (Singer et al., 2006; Gu and Han, 2007; Lamm et al., 2007; Hein and Singer, 2008; Loggia et al., 2008; Cheng et al., 2010; Guo et al., 2012; Reynolds-Losin et al., 2012, 2014, 2015). Many studies have reported concurrent activation of and connectivity between ROI’s within one or more cortical networks associated with affective and cognitive empathy, such as during passive observation of emotions or pain (Christov-Moore and Iacoboni, 2016), passive observation of films depicting personal loss (Raz et al., 2014), reciprocal imitation (Sperduti et al., 2014), tests of empathic accuracy (Zaki et al., 2009), and comprehension of others’ emotions (Spunt and Lieberman, 2013). Co-existence of affective and cognitive mechanisms can be documented even at the level of TMS-induced motor evoked potentials (MEPs), a functional readout of motor excitability (Gordon et al., 2018). A recent study by Christov-Moore et al. (2020) also found that patterns of connectivity between and within cognitive and affective empathy networks was the best predictor of empathic concern, exceeding canionical networks and each network on its own. Thus, the neural instantiation of affective and cognitive empathy may rely on systems that operate like connected clusters in a network, even if they have apparently differentiable spatial instantiations in the brain.

Furthermore, there are additional studies examining the relationship between affective/somatomotor processing and prosocial behavior that are not mentioned in the introduction, such as: non- strategic generosity in the dictator game: Christov-Moore and Iacoboni, 2016; harm aversion in moral dilemmas: Christov- Moore et al., 2017; donations to reduce pain in another: Gallo et al., 2018; helping behavior: Hein et al., 2011; Masten et al., 2011; charitable donations: Ma et al., 2011.

On a final note, though this does not affect my evaluation of the manuscript, I'm curious why the location and name of the university where the study was redacted within the text.

6. PLOS authors have the option to publish the peer review history of their article (what does this mean?). If published, this will include your full peer review and any attached files.

Reviewer #1: No

Reviewer #2: No

---

## [Author Response · Author response to Decision Letter 0]

4 Oct 2021

PONE-D-21-15135

The influence of role uncertainty, empathy induction and trait empathy on dictator game giving

We wish to thank the reviewers for their insightful comments. We have followed their suggestions in revising the manuscript. Our responses to the reviewers’ comments are explained below, and all the changes we have made are highlighted in the revised manuscript.

Reviewer #1

Reviewer #1: Thank you for inviting me to review “The influence of role uncertainty, empathy induction, and trait empathy on dictator game giving.” Herne et al. take a somewhat exploratory approach to address an interesting question: how state empathy, trait empathy, and role uncertainty influence dictator game giving. The researchers manipulated two factors: role uncertainty (information about one’s role) and empathy induction (written responses as to recipient’s feelings). These factors are well-motivated despite the sparse and discrepant findings in the literature. Additionally, the researchers extended the design and measured how participant-levels of empathy, state and trait, interact with the above factors. The research question is of open interest, the methods appear sound, and the results replicate prior work as well as introduce some novel findings. However, the following points require clarification before publication.

Abstract:

Line 27: “Each participant was randomly assigned into one of four conditions and into one of two roles, before or after decisions were made.” Are there not only four conditions, that include the role type? Please clarify this sentence

Response: 

We clarified this in the abstract. (Lines 26-29)

Introduction:

- Line 183: The researchers state their “main interest lies in detecting an interaction effect between different trait empathy dimensions and manipulations of role uncertainty and empathy induction.” However, the interaction focus is not clearly motivated in the introduction. It reads secondary to the main effects and, in fact, the researchers actually predict an additive role of the factors (lines 175 -177). Please provide more evidence for hypothesizing the interaction, and motivate the rationale earlier in the introduction, if the interaction is indeed the main interest of the study.

- It would be helpful to adopt a slightly broader motivation in the introduction, particularly to motivate the DV. Rather than stating the purpose is to “increase dictator game giving”, it would be helpful to describe what this means. Is dictator game giving an index of prosociality, i.e., at the cost to oneself? Or altruism? A latent operationalization is needed.

- The hypotheses (under “Research Questions”) read a little disorganized and verbose. Please streamline.

Response: 

It is true that our interest in the interaction needs clarification. We added a more thorough consideration of the potential interaction effects in the hypotheses section (lines 212-225). We also added evidence and literature (Guo R, Wu Z. Empathy as a buffer: How empathy moderates the emotional effects on Preschoolers’ sharing. Brit J Psychol. 2021; 112: 412–432. Mayer SV, Jusyte A, Klimecki-Lenz OM, Schönenberg M. Empathy and altruistic behavior in antisocial violent offenders with psychopathic traits. Psychiat Res, 2018; 269: 625–632). Moreover, we clarified our interest in the interaction already in the introduction and tried to give it a better motivation (lines 73-78). 

Regarding the motivation of our dependent variable, we now define and discuss altruistic behavior in more detail in the introduction. Two references are added to this discussion (Batson CD and Powell AA. Altruism and Prosocial Behavior. In: Millon T, Lerner MJ, editors. Handbook of psychology: Personality and social psychology Vol. 5. Hoboken: John Wiley & Sons; 2003, pp. 463–484; Andreoni J. Giving with impure altruism: Applications to charity and Ricardian equivalence. J Polit Econ. 1989; 97: 1447-1458). (Lines 52-66)

The hypotheses section has been streamlined and named “Hypotheses”. (Lines 199-234)

Methods:

- Line 205: include standard deviation

- Line 249: an example in this format- (e.g., XXXX) - for imagine-other and imagine-self tasks would be helpful.

- Please justify the unbalanced design. Why are there more participants in the “uncertain role” manipulation, and specifically, with the empathy induction condition?

Response: 

We included standard deviations. (Lines 240-244)

We added examples of imagine-other and imagine-self tasks. (Lines 286-287)

The unbalanced design is a result of practical necessity: Because each subject under the role uncertainty condition made a decision, those sessions yielded twice the number of observations compared to the treatments with role certainty. As the assignment treatment conditions and session order were randomized, having a complete control over the number of subjects in each cell was not something that would have been easily achieved. Since the homogeneity of variances assumption is not rejected, having unequal cell sizes won’t complicate the statistical testing of the treatment effects. We formulated a justification in the manuscript before Table 1. (Lines 316-319)

Results

- Please specify the total participants included in the analysis, especially when considering all the individual difference measures.

- Line 341: I don’t understand the motivation behind including an effect size difference test, when one main effect is not significant (empathy induction). The rationale of the test should be more clearly motivated, rather than explaining what the analysis does (i.e., significance test from zero, line 343) or that the “effects are the same nor that the effects are different” (line 352).

- Line 368: please specify whether the predictors were standardized or unstandardized

- Line 368: how were the initial predictors in the stepwise regression selected?

- Please add confidence intervals and effect sizes

Response: 

Total number of participants was mentioned under Dictator Game: Of the 186 participants, 131 made an allocation decision – this has now been rewritten to make it more clear. (Lines 266-270)

Line 341: We have clarified our motivation for testing the difference between sizes of the effects of the two manipulations. A reference is also added (Perezgonzalez, J D. Fisher, Neyman-Pearson or NHST? A tutorial for teaching data testing. Front Psychol. 2015; 6). (Lines 389-397)

Line 368: We added ‘unstandardized’ in parenthesis in the description of the model. (Lines 425, 432)

Line 368: We ran a backward stepwise linear regression, which entails a fully saturated model that has each relevant variable included. Based on Akaike’s Information Criteria, the procedure removes variables whose inclusion did not improve the explanatory power of the model. So there was no selection for the initial predictors, with the exception of fantasy and distress IRI subscale measures, which we expected not to be pertinent to the analysis, based on the description of the subscales (cf. “Fantasy – taps respondents' tendencies to transpose themselves imaginatively into the feelings and actions of fictitious characters in books, movies, and plays”, Personal Distress measures "self-oriented" feelings of personal anxiety and unease in tense interpersonal settings” (Davis, M. H. (1983). Measuring individual differences in empathy: Evidence for a multidimensional approach. Journal of Personality and Social Psychology, 44, 113–126). We added a justification for dropping fantasy and distress dimensions and including perspective-taking and empathic concern in the regressions. A reference is added to this discussion (Chrysikou, EG, Thompson WJ. Assessing cognitive and affective empathy through the Interpersonal Reactivity Index: An argument against a two-factor model. Assessment 2016; 23: 769–777). (Lines 426-430)

We added confidence intervals on both initial and final models, and Cohen’s f2 effect sizes for the final models II and IV – including effect sizes in the tables describing the initial models wouldn’t have helped our analysis, as the fully saturated models had several extra explanatory variables that contributed scantily towards explaining the observed behavior, and consequently they were dropped during the stepwise regression. (Tables 5 and 6)

Discussion:

- Line 421 – 423: “role uncertainty was effective only when”- effective on what? Please be explicit.

- Please rephrase the paragraph starting at line 483 to avoid confusability.

- Line 484 does not follow clearly. Please correct to: “within the scale range, role uncertainty had a smaller influence on DG behavior (i.e., your DV) for those…”)

- Lines 485- 492: Interpretation of interaction effect is somewhat speculative, although logical. I would suggest that the authors include some relevant citations.

- Line 469: please add citation after recipient.

Please tone down the language in certain conclusions. For instance, line 497: “those who tend to be concerned by others are not influenced by role uncertainty…”

Line 510: The researchers state: “our results suggest that affective rather than cognitive processes enhance altruistic behavior.” Using the Dictator game as an index for altruistic behavior, is never clearly described nor is the focus on altruism clearly motivated in the introduction. Altruistic behavior should be operationalized and the researchers should explain how and why the dictator game measures this much earlier in the manuscript.

Response: 

Line 421 – 423: The sentence is rephrased to be more explicit. (Lines 482-483)

The paragraph starting at line 483 (in the submitted version) is reformulated. (Lines 548-552)

Line 484 (in the submitted version) is corrected according to the referee’s suggestion. (Lines 549-551)

- Lines 485- 492: We clarified the interpretation of interaction effect and added a citation to further back up our interpretation of the effect (Penner LA, Dovidio JF,. Piliavin JA, Schroeder DA. Prosocial behavior: Multilevel perspectives. Annu Rev Psychol. 2005; 56: 365-392). (Lines 556-569)

- Line 469: We added the requested citation. (Line 543)

We toned down the language in certain conclusions in accordance with the reviewer’s suggestion. (Lines 472-587)

Line 510: Indeed, the meaning of altruism was not clearly stated in the original version of the manuscript. We clarified the meaning of altruism, as well as the relationship between dictator game giving and altruism already in the introduction and tried to give a better motivation for our focus on altruism. (Lines 52-66)

Minor:

Please improve the readability of the manuscript. Some minor suggestions are listed below.

- Line 43: change “charities is” to “charities are”

- Line 79: Get rid of question mark

- Line 83: change “as” to “or”

- Line 123: “public good game”. What is this? Helpful to include an example (e.g., XXXX)

Response: 

We fixed the grammatical errors and typos, and provided a definition and an example, as well as review and meta-analysis references for public good games (Public good game is explained in lines 153-157 and two references are added (Ledyard J. Public Goods: A Survey of Experimental Research: In: J. Kagel J, Roth A, editors. The Handbook of Experimental Economics; 2020. Princeton: Princeton University Press, pp. 111-194. Zelmer J. Linear public goods experiments: A meta-analysis. Exp Econ. 2003;6: 299-310). 

Reviewer #2

Reviewer #2: The authors examined the roles of and interactions between empathy induction and role uncertainty on sharing behavior in an anonymized, one-shot dictator game, and examined the mediating role of trait empathy on these main effects using an across-subject design. By and large this study is well-designed, well-powered, and the grounding theoretical hypothesis is compelling: that role uncertainty promotes a form of perspective-taking (which could be argued to be a more effective, implicit form of empathy induction).

Regarding the empathy induction: While the authors do provide supporting evidence for the format of the empathy induction, they should address the seemingly cognitive nature of the induction, in light of the greater influence of affective empathy on empathic concern (as mentioned in the introduction). Future studies on this subject should perhaps include a more somatomotor and affective empathy induction protocol.

Response: 

This is a relevant point. We added a consideration of the possible differences between affective and cognitive empathy inductions in the discussion. (Lines 581-584)

Regarding the statistical analyses: Did the authors test for normality of offers? Depending on this factor, they may have been better off reporting the median primarily and using non-parametric tests. Simply assuming normality can lead to erroneous conclusions.

Response: 

We performed Two-way ANOVA analysis to analyze the effect of our independent variables on dictator giving. To test the normality and the fit of our model, we performed Levene’s Test for homoscedasticity, which the residuals (i.e. errors) passed (Levene’s Test, F(3,127) = 1.6, p = .19). However the residuals were not normal (Shapiro-Wilk, W = 0.95, p = .0001).

For a 2x2 Anova, testing the residuals of the whole model is equivalent of testing the normality of residuals (errors) separately in each treatment cell (that is, the distribution of the dependent variable conditional on the treatment variable, Y|X). As the reviewer points out, assuming normality - which our residuals aren’t - might sometimes lead to erroneous conclusions. On the other hand, F-test is fairly robust for deviations from normality, and given our cell size, central limit theorem comes to rescue as per usual. Please also note the response below regarding the (un)importance of the normality assumption. 

Also, while the mean and median may be non-significantly different between some of the conditions, there may be significant differences between the shape of the distributions, and vice-versa. I recommend running a Kolmogorov–Smirnov test to ascertain whether this is the case.

Response: 

As such, we are interested in the level effect (or change in mean location) of our treatments, and while interesting, for our design, simple non-parametric tests cannot unfortunately distinguish between different shapes of distributions under different combinations of our treatment conditions while also taking the possible interaction into account at the same time. This is also why we keep within the ANOVA framework and do not test nor report non-parametric tests on the response variables in the manuscript. 

Furthermore, a well-known statistical authority goes so far as to say that normality is the least important assumption of linear models, and barely important at all, and even explicitly recommends against running diagnostics on residuals (Gelman, A., & Hill, J. (2006). Data analysis using regression and multilevel/hierarchical models. Cambridge university press).

Regarding the background/introduction, the authors present evidence that cognitive and affective empathy are subserved by distinct systems. While lesion and activation studies do support this interpretation, the issue may be more complex than presented: the neural bases of affective and cognitive empathy processes interact considerably. Recent research suggests that somatomotor and affective processing contribute to our evaluations of others’ internal states, beliefs, and intentions (Gallese, 2007; Schulte-Rüther et al., 2007; Frith and Singer, 2008; Obhi, 2012; Christov-Moore and Iacoboni, 2016; Christov-Moore et al., 2017a), as well as our decisions about others’ welfare (Greene, 2001; Camerer, 2003; Van’t Wout et al., 2006; Oullier and Basso, 2010; Hewig et al., 2011; Christov- Moore et al., 2017). Conversely, cognitive processes are increasingly implicated in the contextual modulation of neural resonance, a putative substrate of affective empathy (Singer et al., 2006; Gu and Han, 2007; Lamm et al., 2007; Hein and Singer, 2008; Loggia et al., 2008; Cheng et al., 2010; Guo et al., 2012; Reynolds-Losin et al., 2012, 2014, 2015). Many studies have reported concurrent activation of and connectivity between ROI’s within one or more cortical networks associated with affective and cognitive empathy, such as during passive observation of emotions or pain (Christov-Moore and Iacoboni, 2016), passive observation of films depicting personal loss (Raz et al., 2014), reciprocal imitation (Sperduti et al., 2014), tests of empathic accuracy (Zaki et al., 2009), and comprehension of others’ emotions (Spunt and Lieberman, 2013). Co-existence of affective and cognitive mechanisms can be documented even at the level of TMS-induced motor evoked potentials (MEPs), a functional readout of motor excitability (Gordon et al., 2018). A recent study by Christov-Moore et al. (2020) also found that patterns of connectivity between and within cognitive and affective empathy networks was the best predictor of empathic concern, exceeding canionical networks and each network on its own. Thus, the neural instantiation of affective and cognitive empathy may rely on systems that operate like connected clusters in a network, even if they have apparently differentiable spatial instantiations in the brain.

Response: 

We thank the reviewer for this thorough comment. Considering that the present manuscript is not about the neural bases of affective and cognitive empathy we feel that we cannot go into this matter very deeply, here. But we definitely agree with the reviewer in that our single sentence on this topic gave an imbalanced impression for a reader. In fact, in the sentence preceding this one we wrote that “…empathy as a multidimensional concept comprising distinct but related cognitive and affective processes”. Therefore, we modified this “neural basis” sentence (and the related references) and, in the present version of the manuscript, it reads now: “This view is also supported by brain research showing that cognitive and affective dimensions of empathy are represented by separate but interacting neural networks [16-21]”. We removed some of the references appearing in the previous version of the manuscript and added some new ones to give a reader a more balanced view of the state of knowledge on this field. The references are now:

Christov-Moore L, Iacoboni, M. Self-other resonance, its control and prosocial inclinations: brain-behavior relationships. Hum Brain Mapp 2016; 37: 1544–1558. doi: 10.1002/hbm.23119

Christov-Moore L, Reggente N, Douglas PK, Feusner JD, Iacoboni M. Predicting

empathy from resting state brain connectivity: A multivariate approach.

Front Integr Neurosci 2020; 14:3. doi: 10.3389/fnint.2020.00003

Fan Y, Duncan NW, de Greck M, Northoff G. Is there a core neural network in empathy? An fMRI based quantitative meta-analysis. Neurosci Biobehav Rev 2011; 35: 903–911. doi: 10.1016/j.neubiorev.2010.10.009

Eres R, Decety J, Louis WR, Molenberghs P. Individual differences in local gray matter density are associated with differences in affective and cognitive empathy. Neuroimage 2015; 117: 305–310. doi: 10.1016/j.neuroimage.2015.05.038 

Hillis AE. Inability to empathize: Brain lesions that disrupt sharing and understanding another’s emotions. Brain 2014; 137: 981–997. doi: 10.1093/brain/awt317 

Zaki J, Weber J, Bolger N, Ochsner K. The neural bases of empathic accuracy. Proc Natl Acad Sci USA 2009; 106: 1–6. doi: 10.1073/pnas.0902666106

(Lines 129)

Furthermore, there are additional studies examining the relationship between affective/somatomotor processing and prosocial behavior that are not mentioned in the introduction, such as:: Christov-Moore and Iacoboni, 2016; harm aversion in moral dilemmas: Christov- Moore et al., 2017; donations to reduce pain in another: Gallo et al., 2018; helping behavior: Hein et al., 2011; Masten et al., 2011; charitable donations: Ma et al., 2011.

Response: 

In the present version of the manuscript, we mention those studies which have directly investigated the association between the neural empathy networks and prosocial behavior. Given that earlier in the manuscript, we shortly mention the neural substrates associated with empathy, mentioning these studies particularly seemed appropriate for the present context.

We added a new sentence: “Brain imaging studies have shown that activation of the neural systems associated with affective empathy correlates with the amount of donations in the dictator game [16], donations to a charitable organization [24], and with the degree of providing verbal comfort and support (prosocial behavior) towards socially excluded individuals [25].”

The new references are:

Christov-Moore L, Iacoboni, M. Self-other resonance, its control and prosocial inclinations: brain-behavior relationships. Hum Brain Mapp 2016; 37: 1544–1558. doi: 10.1002/hbm.23119

Ma Y, Wang C, Han S. Neural responses to perceived pain in others predict real-life monetary donations in different socioeconomic contexts. Neuroimage 2011; 57: 1273–1280. doi: 10.1016/j.neuroimage.2011.05.003

Masten CL, Morelli SA, Eisenberger NI. An fMRI investigation of empathy for 'social pain' and subsequent prosocial behavior. Neuroimage 2011; 55: 381–388. doi: 10.1016/j.neuroimage.2010.11.060.

(Lines 137-140)

On a final note, though this does not affect my evaluation of the manuscript, I'm curious why the location and name of the university where the study was redacted within the text.

Response: 

The identification details of the university/experimental laboratory were redacted because the manuscript was originally written for a double-blind procedure in mind. The name of the university is now added. (Line 311)

We also added a citation to the Finnish translation of IRI (Silfver M, Helkama K, Lönnqvist J-E, Verkasalo M. The relation between value priorities and proneness to guilt, shame, and empathy. Motiv Emotion 2008; 32: 69–80). (Line 251)

---

## [Decision Letter · Decision Letter 1]

10 Nov 2021

PONE-D-21-15135R1The influence of role uncertainty, empathy induction and trait empathy on dictator game givingPLOS ONE

Dear Dr. Herne,

Thank you for submitting your manuscript to PLOS ONE. After careful consideration, we feel that it has merit but does not fully meet PLOS ONE’s publication criteria as it currently stands. Therefore, we invite you to submit a further revised version of the manuscript that addresses all the points raised by reviewer 1 during the review process of your revised manuscript.

We look forward to receiving your revised manuscript.

Kind regards,

Marco Iacoboni

Academic Editor

PLOS ONE

Journal Requirements:

Reviewers' comments:

Reviewer's Responses to Questions

**Comments to the Author**

1. If the authors have adequately addressed your comments raised in a previous round of review and you feel that this manuscript is now acceptable for publication, you may indicate that here to bypass the “Comments to the Author” section, enter your conflict of interest statement in the “Confidential to Editor” section, and submit your "Accept" recommendation.

Reviewer #1: (No Response)

Reviewer #2: All comments have been addressed

2. Is the manuscript technically sound, and do the data support the conclusions?

Reviewer #1: Yes

Reviewer #2: Yes

3. Has the statistical analysis been performed appropriately and rigorously? 

Reviewer #1: Yes

Reviewer #2: Yes

4. Have the authors made all data underlying the findings in their manuscript fully available?

Reviewer #1: Yes

Reviewer #2: Yes

5. Is the manuscript presented in an intelligible fashion and written in standard English?

Reviewer #1: Yes

Reviewer #2: Yes

6. Review Comments to the Author

Reviewer #1: Thanks for updating and revising many of the points raised in the prior review. As stated previously, the authors have explored an interesting research question spanning many fields including empathy, prosocial behavior, altruism, and decision-making.

The authors have for the most part done well to address many of the points raised in the previous review. I find their new additions to significantly improve the readability of the manuscript.

However, small areas of clarification remain, mainly in relation to clarity of the research motivation, discussion, and statistical analyses. Many of these were comments regarding clarification I had raised in the previous review, which I still feel the authors can improve. Note that my review is fairly specific in nature, as I do not read any major or new concerns with the paper as it stands. Hence, I detail below a few more minor revisions for the authors before publication:

Please add an interpretation sentence of the results at the end of the abstract

The motivation/hypotheses still read a bit disorganized. One helpful approach I would recommend is to shorten the text in the introduction (~8 pages). This is what I had meant by streamlining rather than just pasting in new text. Clarification would also be helpful. For example, in lines 101 – 106, the authors state “positions in the income distribution are randomly decided after choices are made.” Please elaborate on “positions” (recipient positions?) and make this more accessible to the layreader. It may seem minor, but is a premise and motivation behind your manipulation of the first factor.

Results: Please provide a sample size justification.

In your statistical reports of the analyses, please consider a better name for the role uncertainty factor. I say this because you term one of your levels of the factor, “role uncertainty”, which can get quite confusing for the reader. Maybe consider using “role type” as the name for the factor, and “role certainty” and “role uncertainty” as the two levels (e.g., line 380).

Lines 379 – 382. I do not follow here. The authors state: The results revealed that the main effect of the role uncertainty treatment was highly significant, F(1, 127)= 17.44, p=0.000055. In other words, role uncertainty increased giving over and above the increase that empathy induction caused”

- Why do the authors not report the direction of this main effect, and instead take a regression-like interpretation of the ANOVA?

Minor: I would recommend abbreviating RU and EI at the beginning of the results, and not in the middle (i.e., line 392)?

Starting at line 576, if both empathy induction and role uncertainty can be interpreted as methods for perspective-taking as the authors claim, then a discussion of why empathy induction was not significant here is warranted. I would suggest merging text from the paragraph starting at 586 with this paragraph. They could make the claim of inducing more cognitive-like aspects from empathy induction or the ‘justification’ of behavior aspect driving the lack of significance in the empathy induction manipulation. This would make the discussion also read less disjointed.

It would be helpful if the authors copy and remove some of the text from the paragraph in the discussion section starting at line 490, which would be a great starting point for the paragraph with the effect size test in the results

Reviewer #2: My comments have been adequately addressed. I endorse the publication of this manuscript in its current form. Good work.

7. PLOS authors have the option to publish the peer review history of their article (what does this mean?). If published, this will include your full peer review and any attached files.

Reviewer #1: No

Reviewer #2: No

---

## [Author Response · Author response to Decision Letter 1]

24 Nov 2021

Our responses to reviewers' comments are explained in a separate file.

---

## [Decision Letter · Decision Letter 2]

20 Dec 2021

TThe influence of role awareness, empathy induction and trait empathy on dictator game giving

PONE-D-21-15135R2

Dear Dr. Herne,

We’re pleased to inform you that your manuscript has been judged scientifically suitable for publication and will be formally accepted for publication once it meets all outstanding technical requirements.

Kind regards,

Marco Iacoboni

Academic Editor

PLOS ONE

Additional Editor Comments (optional):

Reviewers' comments:

Reviewer's Responses to Questions

**Comments to the Author**

1. If the authors have adequately addressed your comments raised in a previous round of review and you feel that this manuscript is now acceptable for publication, you may indicate that here to bypass the “Comments to the Author” section, enter your conflict of interest statement in the “Confidential to Editor” section, and submit your "Accept" recommendation.

Reviewer #1: All comments have been addressed

2. Is the manuscript technically sound, and do the data support the conclusions?

Reviewer #1: Yes

3. Has the statistical analysis been performed appropriately and rigorously? 

Reviewer #1: Yes

4. Have the authors made all data underlying the findings in their manuscript fully available?

Reviewer #1: Yes

5. Is the manuscript presented in an intelligible fashion and written in standard English?

Reviewer #1: Yes

6. Review Comments to the Author

Reviewer #1: The authors have sufficiently addressed all the prior comments in their revision. Would recommend for publication.

7. PLOS authors have the option to publish the peer review history of their article (what does this mean?). If published, this will include your full peer review and any attached files.

Reviewer #1: No

---

## [Editor Report · Acceptance letter]

14 Jan 2022

PONE-D-21-15135R2 

The influence of role awareness, empathy induction and trait empathy on dictator game giving 

Dear Dr. Herne:

I'm pleased to inform you that your manuscript has been deemed suitable for publication in PLOS ONE. Congratulations! Your manuscript is now with our production department. 

Kind regards, 

on behalf of

Dr. Marco Iacoboni 

Academic Editor

PLOS ONE